# Nurses Working in Nursing Homes: A Mediation Model for Work Engagement Based on Job Demands-Resources Theory

**DOI:** 10.3390/healthcare9030316

**Published:** 2021-03-12

**Authors:** Yukari Hara, Kyoko Asakura, Shoko Sugiyama, Nozomu Takada, Yoshimi Ito, Yoko Nihei

**Affiliations:** 1Graduate School of Medicine, Tohoku University, 2-1 Seiryo-Machi, Aoba-ku, Sendai, Miyagi 980-8575, Japan; asakura@med.tohoku.ac.jp (K.A.); s.sugiyama@med.tohoku.ac.jp (S.S.); takada@med.tohoku.ac.jp (N.T.); y_itoh@med.tohoku.ac.jp (Y.I.); yonihei@med.tohoku.ac.jp (Y.N.); 2School of Nursing, Miyagi University, 1-1 Gakuen, Taiwa-cho, Kurokawa-gun, Miyagi 981-3298, Japan; 3Department of Nursing, Faculty of Health Sciences, Tohoku Fukushi University, 1-8-1 Kunimi, Aoba-ku, Sendai, Miyagi 981-8522, Japan

**Keywords:** nursing home, work engagement, mediation model, job demands-resources theory, nurse, occupational commitment, professional autonomy

## Abstract

This study examined the impact that the attractiveness of working in nursing homes and autonomous clinical judgment have on affective occupational commitment, and whether work engagement mediates these relationships. This analysis was based on the job demands-resources theory. The study setting was 1200 nursing homes (including long-term care welfare facilities and long-term care health facilities) in eastern Japan. An anonymous, self-report questionnaire survey was administered to two nurses from each facility, resulting in a prospective sample of 2400 participants. Overall, 552 questionnaires were analyzed, in which structural equation modeling and mediation analysis using the bootstrap method were performed. The results showed that the attractiveness of working in nursing homes does not directly affect affective occupational commitment; work engagement fully mediates the impact of attractiveness of working in nursing homes on affective occupational commitment. Additionally, autonomous clinical judgment showed a direct impact on both work engagement and affective occupational commitment, indicating that work engagement partially mediates the impact on affective occupational commitment. To increase the affective occupational commitment of nurses working in nursing homes, managers should help nurses recognize the attractiveness of working in nursing homes, and then provide appropriate support to help such nurses work in a motivated manner.

## 1. Introduction

The aging and decline in size of the nursing workforce is a global issue [1,2,3,4]; this especially applies to the older-adult care sector, which is composed of nursing homes and home-based care services [5]. Older adults have a higher incidence of disability and a greater need for health care when compared to younger populations [6] and, as global populations continue to age, the demand for nurses in the older-adult care sector is increasing [7,8]. In attempts to address the high turnover rate and rising demand for nurses in the older-adult care sector, several studies have sought to investigate and clarify the factors that influence the turnover intention and job satisfaction of such nurses (e.g., [9,10]).

Work engagement can reduce nurses’ turnover intention [11,12] and, concurrently, improve their job performance, such as the quality of the care they provide [13]. Therefore, to retain more nurses in the older-adult care sector while maintaining the quality of the care provided, it is necessary to identify the factors that can enhance such nurses’ work engagement. The present study uses the job demands-resources (JD-R) theory [14,15] to explore methods of improving the “motivational process” of nurses working in nursing homes.

JD-R theory integrates two very different processes, namely, a health-impairment process and a motivational process [14,15]. The health-impairment process occurs in response to job demands and can result in negative outcomes such as burnout and, ultimately, job turnover. Meanwhile, the motivational process, which is focused on in this study, occurs when job and personal resources increase work engagement; this can lead to positive work outcomes such as high work performance [14,15]. JD-R theory emphasizes that, in academic research, selection of specific job demands and resources for analysis should be based on consideration of the profession in question and the associated workplace settings [14]. The number of empirical studies in which JD-R theory is used in attempts to improve nurses’ well-being is increasing, but most of these studies have been conducted in hospital settings (e.g., [11,16]). The aging of the global population has resulted in an increased demand for nurses in the older-adult care sector [7,8]. In order to retain more such nurses while maintaining a high quality of care, research to explore the motivational process of nurses working in nursing homes is needed. It is therefore necessary for researchers in this field to select variables in the context of nurses working in nursing homes.

Several researchers, with the aim of finding methods of retaining existing nurses and attracting new nurses in the older-adult care sector, have focused on the attractiveness of working in the older-adult care sector [17,18,19]. These studies have shown that the older-adult care sector has several attractive features that are not present in hospital work. These include the opportunity to gain in-depth knowledge of patients, to develop long-term relationships with them, and the ability to make independent nursing and clinical judgments and work autonomously.

Nonetheless, despite the attention paid to the attractiveness of working in the older-adult care sector, the impact of the motivational process in improving the well-being of nurses working in nursing homes has not been examined. To the best of our knowledge, this is the first study to consider the attractiveness of working in nursing homes as a job resource that influences the motivational process outlined in JD-R theory. In JD-R theory, job resources refer to the physical, psychological, social, and organizational aspects of a job that help workers achieve work goals; that reduce workers’ job demands and the associated physiological and psychological costs; or that stimulate workers’ personal growth, learning, and development [14,15,20]. In the attractive work model [21], work attractiveness is defined as a positive characteristic of work. High work attractiveness means an organization is perceived as a positive workplace, and such positive perceptions allow employees to experience work stability as well as promoting identification and dedication [21]. This explains why work attractiveness is a job resource. In addition, work attractiveness has been shown to have a significant positive correlation with work engagement [22]. Work engagement is defined as a positive, fulfilling, and affective-motivational state of work-related well-being characterized by vigor, dedication, and absorption [23]. Affective occupational commitment, on the other hand, is the affective bond between an employee and their organization [24]. The present study examines the impact the attractiveness of working in nursing homes has on affective occupational commitment, and whether work engagement mediates this relationship. The resultant findings may help solve the workforce problems experienced by nursing homes by suggesting approaches through which existing nurses can be retained and new nurses recruited.

This study treats autonomous clinical judgment as a personal resource in the context of nursing homes. Personal resources are the beliefs people hold regarding the level of control they have over their environment [14,15,20]. Professional autonomy, of which autonomous clinical judgment is a constituent factor, is defined, for nurses, as the right to make independent nursing and clinical judgments and exert control over their nursing practice [25]. Consequently, it is recognized as a professional resource in the nursing context [13]. Professional autonomy has been found to have a significant positive effect on work engagement [26,27]. Thus, autonomous clinical judgment is a core element of nurses’ professional autonomy [25].

The percentage of registered nurses who provide direct care in nursing homes, varies across countries, but is generally small (e.g., [28,29]). For example, in Japan there are two types of long-term care facilities for older adults who do not require hospitalization: long-term care welfare facilities and long-term care health facilities [30]. In these facilities, the number of nurses (including registered nurses and licensed practical nurses) per 100 capacity is 11 and five, respectively [31]. As a result of the small number of medical professionals assigned, it is very important for nurses to have the ability to make autonomous clinical judgments. To examine this, in the present study we measured, among a sample of such nurses, attitudes toward autonomous clinical judgment. This was performed using the Attitude Toward Professional Autonomy Scale for Nurses [25]; this scale can be used even in situations where it is difficult for nurses to act autonomously, as is the case in Japan [25].

The small number of medical professionals in nursing homes has created a serious situation. In providing care for older adults, who can have multiple comorbidities and are characterized by a need for long-term care, deficiencies in care capabilities and decision-making staff increases time and performance pressure, resulting in chronic physical stress for nurses [32]. Notably, nurses working in the older-adult care sector have been found to be more aware of workforce shortages and inadequate skillsets than are nurses who work in the acute sector [33]. If, through the course of this research, it is determined that, in the context of nursing homes, nurses’ attitudes toward autonomous clinical judgment influences affective occupational commitment through the mediation of work engagement, this may contribute to the development of strategies to solve these issues.

Furthermore, there is some evidence that the attractiveness of working in nursing homes and autonomous clinical judgment impact affective occupational commitment through the mediation of work engagement. Work engagement has been shown to have a significant positive association with affective occupational commitment [34,35]. In the motivational process of JD-R theory, it has been demonstrated that work engagement mediates the relationship between job and personal resources and affective occupational commitment [36]. In addition, the attractive work model [21] emphasizes that work attractiveness is a concept that influences work engagement and occupational commitment, respectively [21]. This shows that the attractiveness of working in nursing homes has a direct impact on affective occupational commitment, and that work engagement may partially mediate this relationship. Further, although based on less evidence than that concerning affective organizational commitment, it has been suggested that professional autonomy also has a direct impact on affective occupational commitment [37]. Professional autonomy has been shown to have a significant positive impact on work engagement [26,27], indicating that work engagement may partially mediate the relationship between autonomous clinical judgment, which is a constituent factor of professional autonomy, and affective occupational commitment.

Considering the above, the present study aimed to explore the impact that the attractiveness of working in nursing homes and autonomous clinical judgment have on affective occupational commitment, and, to determine whether work engagement mediates these relationships. By clarifying the mechanism by which these job resources and personal resources affect the improvement of nurses’ well-being, it contributes to the improvement of well-being of nurses working in nursing homes, and it could be used to identify means to retain more nurses in the older-adult care sector while also maintaining a high quality of care provided to older adults. In addition, by exploring the motivational processes of nurses working in nursing homes, the findings could add to existing nursing-focused research based on JD-R theory, providing new insight regarding the motivational process in the context of nursing homes. Thus, this research contributes to the extension of JD-R theory and allows for a more detailed interpretation of the theory.

Based on the above literature review, we have developed the following two hypotheses. Figure 1 shows the hypothetical model developed for this study of the motivational process for nurses working in nursing homes.

Hypothesis 1: The attractiveness of working in nursing homes has a direct positive impact on affective occupational commitment, and work engagement partially mediates this relationship.

Hypothesis 2: Autonomous clinical judgment has a direct impact on affective occupational commitment, and work engagement partially mediates this relationship.

**Figure 1 healthcare-09-00316-f001:**
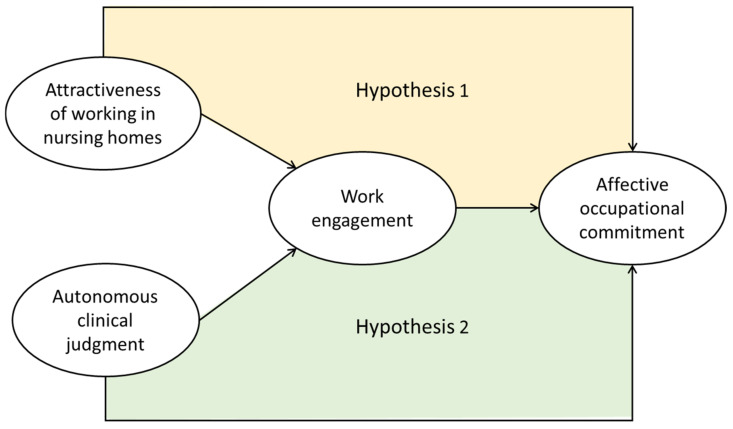
Motivational process model for nurses working in nursing homes.

## 2. Materials and Methods

In calculating the sample size, the required number of participants was calculated so that the average of the scores of “autonomous clinical judgment” would not be too different from the average of the population. In a previous study conducted by this research team, the standard deviation of the total scale score was 12 [38]. Additionally, assuming that the error is 1 and the reliability is 95% in this survey, the required number of participants is 556 when calculated using the sample size calculation formula. In this study, the questionnaire was collected by the mail method, so assuming that the collection rate of the questionnaire is 25%, the required number of participants is 2224. The number of target facilities was 1200 because we estimated a little more and decided to target a total of 2400 nurses, two at each facility. Thus, 1200 facilities were randomly selected from 6055 nursing homes in eastern Japan as the settings for this research.

In this study, nursing homes refer to long-term care welfare facilities and long-term care health facilities. In February 2019, we distributed an anonymous, self-report questionnaire survey to a total of 2400 nurses from these facilities (both registered nurses and licensed practical nurses); that is, to two nurses from each facility. The questionnaires were mailed to facility managers, who then distributed them to nurses. Completed questionnaires were returned to the researchers by mail. The survey period was February to May 2019. The questionnaire contained information regarding nurses’ demographics, the attractiveness of working in nursing homes, autonomous clinical judgment, affective occupational commitment, and a shortened version of the Japanese version of the Utrecht Work Engagement Scale (UWES).

### 2.1. Ethical Considerations

The ethics committee at Tohoku University School of Medicine granted approval for this study (no. 2018-1-717) on 17 January 2018. Participants were assured of their confidentiality and anonymity during both the research and publication process. Prior to their participation, we informed participants of the purpose and design of the study, and that participation was voluntary. Returned/submitted questionnaires were deemed to represent consent to participate. This study conformed to the provisions of the Declaration of Helsinki in 1995 (as revised in Edinburgh 2000).

### 2.2. Measures

#### 2.2.1. Demographic Variables

Data on individual and work-related variables were collected: sex, age, marital and child status, educational background, years of nursing experience, years of experience at the current facility, and work position.

#### 2.2.2. Attractiveness of Working in Nursing Homes

By referring to previous research regarding the attractiveness of working in the older-adult care sector [17,18,19], we created original question items that measured the degree of attractiveness of working in nursing homes. The question items were as follows: “The nursing work performed at nursing homes has a different appeal to that performed at hospitals.” “I find working closely with patients in regard to all aspects of their lives, as is the case when working at a nursing home, attractive.” “Working in nursing homes is rewarding because nurses are often required to make autonomous decisions.” Responses were provided using a five-point Likert-type scale, ranging from 1 (“strongly disagree”) to 5 (“strongly agree”); higher scores indicated that the respondent was strongly attracted to working in nursing homes. After exploratory factor analysis, which showed a single-factor structure, we performed confirmatory factor analysis, which confirmed the validity of the three items and the single-factor structure. The Cronbach’s α coefficient for this study was 0.84.

#### 2.2.3. Autonomous Clinical Judgment

We measured autonomous clinical judgment using the Attitude Toward Professional Autonomy Scale for Nurses developed by Asakura et al. [25] to measure nurses’ attitudes toward professional autonomy; autonomous clinical judgment represents one of the three subscales of this scale. The subscale comprises seven items, including the item, “I want to use my own judgment when practicing nursing.” Responses are given using a five-point Likert-type scale, ranging from 1 (“strongly disagree”) to 5 (“strongly agree”). Higher scores indicate that the respondent has a progressive attitude toward professional autonomy. The Cronbach’s α coefficient for this study was 0.82.

#### 2.2.4. Affective Occupational Commitment

To assess affective occupational commitment, the Japanese version of the Allen and Meyer Three-Dimensional Commitment Scale, which was translated into Japanese and validated by Satoh, Asakura, Watanabe, and Shimojo [39] was used. The scale comprises six question items, including the item, “nursing is important for my self-image.” Responses are given using a five-point Likert-type scale, ranging from 1 (“strongly disagree”) to 5 (“strongly agree”), with higher scores indicating higher occupational commitment. The Cronbach’s α coefficient for this study was 0.85.

#### 2.2.5. Work Engagement

Work engagement was measured using the Japanese version of the UWES (Shimazu Laboratory, Tokyo, Japan) [40]. This scale comprises nine items; three items concerning vigor (e.g., “at my work, I feel bursting with energy”), three concerning dedication (e.g., “I am enthusiastic about my job”), and three concerning absorption (e.g., “I get carried away when I’m working”). All items were rated using a seven-point Likert-type scale, ranging from 0 (“none”) to 6 (“always”). The Cronbach’s α coefficient for this study was 0.93.

### 2.3. Data Analyses

Descriptive statistics were calculated for individual and work-related attributes. Further, factor analysis was conducted on all scales to ensure the original structure of the scale was reproduced. We also checked the distribution and mean of the scores for all scales. Internal consistency tests (Cronbach’s α) were conducted on all scales. Then, the correlation coefficients between scores for each scale were calculated. To evaluate the research hypotheses, structural equation modeling was performed using IBM SPSS v26 and IBM AMOS v26 (SPSS Inc., Chicago, IL, USA). For the mediation model of work engagement, the significance of the indirect effect was examined through mediation analysis using the bootstrap method (with 2000 resamples).

## 3. Results

A total of 571 questionnaires were collected, of which 19 were excluded due to missing data for individual and work-related variables in each scale; finally, 552 completed questionnaires were used for the analysis (valid response rate: 23.0%). The participants’ characteristics are shown in Table 1.

Table 2 shows the mean and standard deviation for each study variable, as well as the correlation coefficients between each of these variables.

The final model for hypothesis testing is shown in Figure 2. The goodness-of-fit index for the final model shows that this model had a statistically good goodness-of-fit (chi-square/degree of freedom ratio = 2.65, goodness-of-fit index = 0.91, adjusted goodness-of-fit index = 0.89, comparative fit index = 0.95, root mean square error of approximation = 0.05, Akaike information criterion = 816.56).

There was no direct effect between attractiveness of working in nursing homes and affective occupational commitment. There was a direct and significant positive effect between attractiveness of working in nursing homes and work engagement (β = 0.48, *p* < 0.001), and a direct and significant positive effect between work engagement and affective occupational commitment (β = 0.46, *p* < 0.001). Thus, part of Hypothesis 1 was not supported, because work engagement fully (rather than partially) mediated the relationship between attractiveness of working in nursing homes and affective occupational commitment (effect = 0.11, standard error (SE) = 0.02, 95% CI (0.06, 0.15), *p* < 0.001).

In addition, there was a direct and significant positive effect between autonomous clinical judgment and work engagement (β = 0.23, *p* < 0.001), and between autonomous clinical judgment and affective occupational commitment (β = 0.21, *p* < 0.001). Therefore, the relationship between autonomous clinical judgment and affective occupational commitment was determined to be partially mediated by work engagement (effect = 0.22, SE = 0.03, 95% CI (0.16, 0.28), *p* < 0.001). Consequently, Hypothesis 2 was supported. Overall, this model described 34% of the variance in work engagement and 33% of the variance in affective occupational commitment.

## 4. Discussion

Previous nursing-based studies involving JD-R theory have focused primarily on nurses in hospital settings; little attention has been paid to the situation in nursing homes. This study aimed to bridge this gap by examining the nursing-home context using the variables associated with the motivational process described in JD-R theory. Specifically, we examined the respective effects attractiveness of working in nursing homes and autonomous clinical judgment have on affective occupational commitment, and whether work engagement mediates these relationships. The results showed that both attractiveness of working in nursing homes and autonomous clinical judgment influence affective occupational commitment through the mediation of work engagement (fully for the former and partially for the latter); this model showed statistically appropriate goodness-of-fit. This result can add new insights regarding the motivational process in the context of nurses who work in nursing homes. Further, the findings not only contribute to the expansion of JD-R theory but can also be used to develop approaches for improving the well-being of nurses working in nursing homes. In particular, such approaches have the potential to improve nurse retention and the recruiting of new nurses in the older-adult care sector.

In this study, it was speculated that the attractiveness of working in a nursing home may act as a job resource, thus affecting work engagement and affective occupational commitment. However, we found that the attractiveness of working in nursing homes does not have a direct effect on affective occupational commitment. This contradicts the existing claim in the attractive work model [21] that work attractiveness is a concept that influences occupational commitment; moreover, it does not support part of this study’s Hypothesis 1. This is a novel discovery of this research, as it reveals the mechanism by which the attractiveness of working in nursing homes influences affective occupational commitment. In other words, this result shows that perceiving working in nursing homes as attractive is not independently sufficient to induce an increase in affective occupational commitment; instead, this factor only has an impact when work engagement increases, leading to an increase in affective occupational commitment. Increasing affective occupational commitment is important, as it has been shown to reduce nurses’ turnover intention [38,41] and to increase job satisfaction and professional competence [42]. For managers, to increase the affective occupational commitment of nurses working in nursing homes, it is necessary to help nurses recognize the attractiveness of working in nursing homes and to provide appropriate support so that they can work in a motivated manner. In addition, some hospital nurses are attracted to nursing in the older-adult care sector [19]. By actively promoting the attractiveness of working in nursing homes to nurses working in other sectors (such as hospital nurses), managers may be able to recruit new nurses. Recruiting and empowering nurses who are attracted to the older-adult care sector could lead to higher retention rates and solve staff shortages. This would be an important result, as the demand for nurses in the older-adult care sector is increasing as the world population ages [7,8].

Furthermore, we found autonomous clinical judgment to have a direct positive effect on both work engagement and affective occupational commitment, and the indirect effect of autonomous clinical judgment on affective occupational commitment through the mediation of work engagement was also found to be significant. This supported this study’s Hypothesis 2, and suggests that nurses with a progressive attitude toward autonomous clinical judgment increase their work engagement and, concurrently, increase their affective occupational commitment. This accords with self-determination theory [43], which states that motivation to make autonomous clinical decisions leads to high intrinsic motivation, work engagement, and affective occupational commitment. Considering this finding, we suggest that managers foster nurses’ intrinsic motivation by encouraging them to make clinical decisions, rather than strictly controlling and directing their work as this can lead to improved work engagement and affective occupational commitment. In addition, supportive management, education, and experience have been identified as important factors in increasing nurse autonomy [44,45]. It is suggested that not only should the managers provide supportive management, but the nurses working in the nursing home should actively receive education such as training and experience in a variety of cases to enhance their autonomy and, as a result, improve work engagement and affective occupational commitment.

This study has several limitations. First, because our study was conducted in Japan, the findings regarding nursing-home settings are limited to the Japanese context, which may consequently limit the generalizability of the results. Although, in many countries, the proportion of registered nurses working in nursing homes is small (e.g., [28,29]), the number of registered nurses, licensed practical nurses, certified nursing assistants, and care workers varies across countries. These differences are influenced by the laws and the provisions of the long-term care insurance systems present in each country where the salaries and benefits of nurses may also differ. Therefore, it is necessary to consider this background when interpreting the results. Second, this survey was conducted using a self-report questionnaire. Participants may have given more socially acceptable answers than their actual considerations, and sometimes may have been unable to accurately assess themselves. Third, this study features a cross-sectional study design, in which the independent and dependent variables were reported simultaneously by the participants; thus, it was not possible to discuss the temporal relationship between the independent and dependent variables. Longitudinal or experimental studies are required to determine the exact causal relationship.

### Implications for Managers

To increase the affective occupational commitment of nurses working in nursing homes, it may be helpful to first help nurses recognize the attractiveness of working in nursing homes, and then provide appropriate support to help them work in a motivated manner. In addition, we suggest that managers foster nurses’ intrinsic motivation by encouraging them to make clinical decisions, rather than strictly controlling and directing their work as this can lead to improved work engagement and affective occupational commitment.

## 5. Conclusions

The results show that the attractiveness of working in nursing homes does not directly affect affective occupational commitment, and that work engagement fully mediates the impact attractiveness has on working in nursing homes on affective occupational commitment. In addition, autonomous clinical judgment has a direct impact on both work engagement and affective occupational commitment, indicating that work engagement partially mediates autonomous clinical judgment’s impact on affective occupational commitment.

## Figures and Tables

**Figure 2 healthcare-09-00316-f002:**
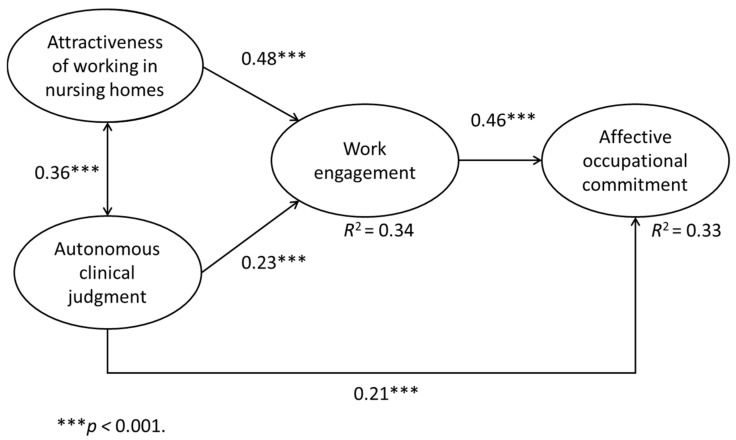
Final motivational process model for this study.

**Table 1 healthcare-09-00316-t001:** Participants’ basic attributes (N = 552).

Variables	N (%)
Sex	
Male	44 (8.0)
Female	508 (92.0)
Mean age (SD †) in years	48.7 (±9.6)
Mean years of nursing experience (SD†)	23.5 (±10.3)
Mean years working at the current facility (SD†)	7.8 (±6.3)
Marital status	
Married	416 (75.4)
Single	97 (17.6)
Divorced or bereaved	39 (7.1)
Number of children	
0	105 (19.0)
1+	447 (81.0)
Educational background	
Vocational school or junior college for registered nurses	532 (96.4)
Baccalaureate program (4-year program in nursing) or masters’ program in nursing	20 (3.6)
Types of nurse licenses	
Registered nurses	410 (74.3)
Licensed practical nurses	142 (25.7)
Position	
Deputy head nurse, head nurse, deputy director of nursing, and director of nursing	239 (43.4)
Regular nurse	313 (56.7)

† SD: Standard deviation.

**Table 2 healthcare-09-00316-t002:** Means and standard deviations for each study variable, and the correlation coefficients between these variables.

Variables	Mean	SD	1	2	3	4
1 Attractiveness of working in nursing homes	11.53	2.41	-	0.311 ***	0.502 ***	0.375 ***
2 Autonomous clinical judgment	24.86	4.43		-	0.340 ***	0.371 ***
3 Work engagement	26.86	8.77			-	0.596 ***
4 Affective occupational commitment	23.02	4.03				-

*** *p* < 0.001; SD: Standard deviation.

## Data Availability

The datasets used during the current study are available from the corresponding author and can be accessed on reasonable request.

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
