# Peer review of "Nurses Working in Nursing Homes: A Mediation Model for Work Engagement Based on Job Demands-Resources Theory"

_healthcare, 2021, doi:10.3390/healthcare9030316_

Round 1

Reviewer 1 Report

This topic is significant within healthcare. Its not a unique topic as of recently, but due to the region of the research it seems to be original work. 

I would make one recommendation to include some of Bowen's Family System's Theory on how it can be applied to staff retention among nurses in working in nursing homes.

Author Response

Response to Reviewer 1 Comments

Point 1: This topic is significant within healthcare. Its not a unique topic as of recently, but due to the region of the research it seems to be original work.

I would make one recommendation to include some of Bowen's Family System's Theory on how it can be applied to staff retention among nurses in working in nursing homes.

Response 1: Thank you for your helpful advice. We have read Bowen’s Family Systems Theory and discussed it together. Bowen’s Family Systems Theory is a broad theory, and as you suggested, some of the eight concepts may be adaptable to the retention of nurses working in nursing homes. However, while the higher goal is to improve the retention of nurses working in nursing homes, the proximal purpose of this study is to improve the well-being of nurses working in nursing homes. We have determined that the introduction of Bowen’s Family Systems Theory can obscure and mislead the focus of this study. Therefore, Bowen’s Family Systems Theory was not included in the manuscript.

Reviewer 2 Report

Dear Authors,

You must revise the spelling and punctuation (e.g., lines 52, 78, 120, 254, ...)

I consider that the introduction is too extensive and that later, in the discussion section, the same statements are repetead. I would try to be more specific in the introduction and avoid repetition of ideas in different sections of the manuscript (e. g., lines 302-304).

I particularly appreciate the section "limitations" where the Authors reflect on the generalization of the results. 

Research Conclusions should be separated from practice implications implications in the last section.

Kind regards,

Author Response

Response to Reviewer 2 Comments

Point 1: You must revise the spelling and punctuation (e.g., lines 52, 78, 120, 254, ...)

Response 1: Thank you for your helpful advice. We are not native English speakers, and hence we have hired a professional English proofreading service to check the manuscript. According to the English editor, there is no error in lines 52 and 78, except perhaps for the automatic hyphenation/cutting of the words “analysis” (“anal-ysis”) in line 52 and “positive” (“pos-itive”) in line 78 due to formatting. To fix the problem in line 52, we have inserted spaces in between words so that the word “analysis” is not cut. As regards line 78, the issue was automatically fixed when we inserted some sentences along lines 67–71. Additionally, we have fixed all other instances of awkward hyphenation throughout the manuscript.

As regards the typographical error in line 120 and the grammatical error in line 254, we have fixed them as well. However, please note that, because of revisions in the manuscript, the sentence originally in lines 119–120 is now in lines 85–86, while the original sentence in lines 254–255 is now in lines 256–257.   

Thank you so much for pointing these issues out.

Point 2: I consider that the introduction is too extensive and that later, in the discussion section, the same statements are repetead. I would try to be more specific in the introduction and avoid repetition of ideas in different sections of the manuscript (e. g., lines 302-304).

Response 2: Thank you for pointing this out. We avoided repeating statements by adding a description to the relevant part of the introduction (lines 69–71) and modifying one sentence of the discussion (lines 303–305).

Point 3: I particularly appreciate the section "limitations" where the Authors reflect on the generalization of the results.

Response 3: Thank you for your supportive comments.

Point 4: Research Conclusions should be separated from practice implications implications in the last section.

Response 4: Following your advice, we have separated the conclusion and practical implications sections and added them to the discussion section as “4.2. Implications for managers” (line 360).

Reviewer 3 Report

Thank you for the opportunity to review the manuscript.  This is a timely study as the attractiveness of working with elder care in long term care facilities is a topic in need of much research.  

The title of the manuscript is a bit too long.  I recommend trimming it by removing "Exploring the Motivational Process of Nurses..."

Line item 67 states that one of the benefits of working in long-term care is autonomy "..as independent medical professional".  could this statement be misleading?  Do nurses work independently as medical professionals in Japan?  Should this statement clarify that it is independent nursing clinical judgment? 

Line 79.   I am not sure if the authors provided a definition for "work engagement" since it is one of the variables measured.  I was not able to see it. 

Line 88.   Again, clarify that it is independent nursing practice decisions. 

Line 101.  What is the name of the scale used to measure nurses' attitudes to professional autonomy? 

Line 162.  In a previous study conducted by the same authors, the discussion regarding the standard deviation of the scale is not properly cited.  I recommend properly citing as not to be perceived plagiarized. 

Recommend adding a table with the demographic statistics from the sample, such as level of education, salaries, age etc.  I think it will be helpful to the reader and enhance the understanding of the study sample.

Recommend adding to the limitations of the study the nature of the self-report scales.  

Overall this is a great study and well conducted. 

Author Response

Response to Reviewer 3 Comments

Point 1: Thank you for the opportunity to review the manuscript.  This is a timely study as the attractiveness of working with elder care in long term care facilities is a topic in need of much research. 

The title of the manuscript is a bit too long.  I recommend trimming it by removing "Exploring the Motivational Process of Nurses..."

Response 1: Thank you for your supportive comments. Based on your advice, we decided to remove “Exploring the Motivational Process of” from the title. Thus, our revised title is: “Nurses Working in Nursing Homes: A Mediation Model for Work Engagement Based on Job Demands-Resources Theory.”

Point 2: Line item 67 states that one of the benefits of working in long-term care is autonomy "..as independent medical professional".  could this statement be misleading?  Do nurses work independently as medical professionals in Japan?  Should this statement clarify that it is independent nursing clinical judgment?

Response 2: As you pointed out, we agree that this expression could be misleading if nurses make independent medical decisions. We have therefore modified the phrase using the expression “independent nursing and clinical judgment” (lines 67–68).

Point 3: Line 79.   I am not sure if the authors provided a definition for "work engagement" since it is one of the variables measured.  I was not able to see it.

Response 3: The definition of “work engagement” was included in the third to last paragraph of the introduction (beginning line 117 in the original version of the manuscript). We have modified the introduction based on your and the other reviewers’ comments. To facilitate an understanding of the concept, we have moved the definition to lines 83–86).

Point 4: Line 88.   Again, clarify that it is independent nursing practice decisions.

Response 4: We have modified the wording according to your advice. This has been moved to line 95.

Point 5: Line 101.  What is the name of the scale used to measure nurses' attitudes to professional autonomy?

Response 5: The name of the scale used was not described, so we have added it (line 109). In addition, we have added it to the relevant section under “2. Materials and Methods” (lines 216–217).

Point 6: Line 162.  In a previous study conducted by the same authors, the discussion regarding the standard deviation of the scale is not properly cited.  I recommend properly citing as not to be perceived plagiarized.

Response 6: Thank you for pointing this out. A citation has been added to the relevant section (line 167).

Point 7: Recommend adding a table with the demographic statistics from the sample, such as level of education, salaries, age etc.  I think it will be helpful to the reader and enhance the understanding of the study sample.

Response 7: According to your suggestion, we have added Table 1 with the demographic statistics of the sample (line 254).

Point 8: Recommend adding to the limitations of the study the nature of the self-report scales. 

Overall this is a great study and well conducted.

Response 8: Thank you for your advice. We have added the nature of the self-report questionnaire to the limitations of this study (lines 352–355).

Reviewer 4 Report

Materials and Methods section

  • -Are 1200 nursing homes selcted randomly or conveniently? If. convenience sampling, please specify tne inclusion criteria.  
  • - Among 571 questionnaires, only 552 were analysed. Please specify why 19 questionnaires are not icluded. Are they completely returned blank?

Author Response

Point 1: Materials and Methods section

-Are 1200 nursing homes selected randomly or conveniently? If. convenience sampling, please specify tne inclusion criteria. 

Response 1: Thank you for pointing this out. As 1,200 facilities were randomly sampled, we added this information to the text (lines 173–174).

Point 2: Results section

- Among 571 questionnaires, only 552 were analysed. Please specify why 19 questionnaires are not icluded. Are they completely returned blank?

Response 2: The 19 excluded questionnaires were those with missing values in the question items related to personal attributes and those with missing values in each scale. These 19 questionnaires were completely excluded from the analysis. We have added a description of these (lines 250–252).
